# META AUXILIARY LABELS WITH CONSTITUENT-BASED TRANSFORMER FOR ASPECT-BASED SENTIMENT ANALYSIS

## ABSTRACT

Aspect based sentiment analysis (ABSA) is a challenging natural language processing task that could benefit from syntactic information. Previous work exploit dependency parses to improve performance on the task, but this requires the existence of good dependency parsers. In this paper, we build a constituent-based transformer for ABSA that can induce constituents without constituent parsers. We also apply meta auxiliary learning to generate labels on edges between tokens, supervised by the objective of the ABSA task. Without input from dependency parsers, our models outperform previous work on three Twitter data sets and match previous work closely on two review data sets.

## 1 INTRODUCTION

Aspect-based Sentiment Analysis (ABSA) is the task of predicting sentiment polarity towards observed aspects in a sentence. Recent work (Bai et al., 2020; Huang & Carley, 2019; Sun et al., 2019; Wang et al., 2020) used syntactic information from dependency parses to achieve new state-of-the-art results on benchmark ABSA data sets. However, these works (i) assumed the existence of good dependency parsers, and (ii) could not further optimize the pre-defined dependency labels for downstream performance of ABSA. Motivated by these limitations, we propose to induce syntactic information with supervision from the ABSA task.

To take syntax into account, we aim to induce the necessary syntactic information for the ABSA task with inductive biases. We first design a Constituent-based Transformer (*ConsTrans*) to group tokens into constituents supervised by the ABSA objective. We argue that the formation of constituents provides a hierarchical structure of the sentence that is suitable for sentiment analysis. For example, in the sentence *"Chinese dumplings in this restaurant taste very good"* with the aspect term *"Chinese dumplings"*, it is important to accurately assign the phrase *"taste very good"* to the aspect.

Next, as seen in Figure 1, even though the dependency graph structures for both sentences are identical, the sentiment towards *"Chelsea"* is positive for the input sentence on the left and negative for the one on the right. Therefore, the type of syntactic relationship between tokens would be useful to identify the sentiment towards the aspect term. Hence, we further extend *ConsTrans* into a Relational Constituent-based Transformer (*RelConsTrans*) to learn relation embeddings between every pair of tokens in the input sentence. We find that simply adding relation embedding fails to outperform *ConsTrans*. Inspired by Liu et al. (2019), we further extend *RelConsTrans* to supervise the relation embedding with an auxiliary label generator (*RelConsTransLG*). In previous work (e.g. Bai et al., 2020; Huang & Carley, 2019), the dependency parser played the role of the auxiliary label generator. However, such dependency parsers were not trained to provide auxiliary labels meant to improve ABSA. *RelConsTransLG* enables us to train the auxiliary label generator alongside the primary task to generate auxiliary labels that could directly enhance the performance of ABSA.

We evaluate our models on five data sets - restaurant and laptop reviews (Pontiki et al., 2014), ACL14 Twitter14 data (Dong et al., 2014), Twitter15 and Twitter17 from a multi-modal ABSA data set (Yu & Jiang, 2019). Compared against previous work which used dependency parsers, our models outperform them on all the Twitter data sets and matched previous work closely on the review data sets even without the use of constituent or dependency parser.

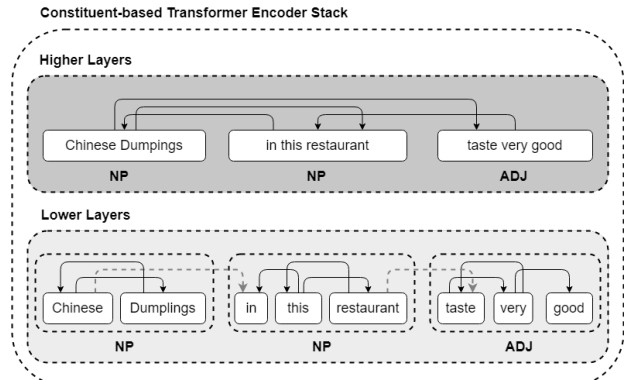

Figure 1: Dependency parse labels as auxiliary labels that help sentiment disambiguation. Tokens in bold and underlined are the aspect terms. Example taken from Bai et al. (2020).

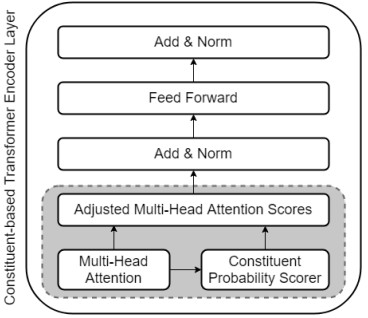

(a) *ConsTrans* Encoder Stack: dotted arrows refer to lower attention weights between tokens from different constituents.

(b) A lower *ConsTrans* layer: the shaded region is different from the vanilla Transformer.

## 2 MODEL FORMULATION

Given a sentence of $m$ tokens, $s = \{w_0, \ldots, w_{m-1}\}$, and a target aspect, $t = \{w_j, \ldots, w_{j+q-1}\}$ of length $q$, the objective of ABSA is to predict the sentiment polarity $y \in \{$negative, neutral, positive$\}$ towards the target aspect $t$ mentioned in sentence $s$. In all our models, we use the pretrained BERT (Devlin et al., 2018) model (BERT-base-uncased) to obtain contextual embeddings as inputs to our model, and we fine tune it together with the model. We format the input to the BERT model as a sentence pair: $[CLS] + s + [SEP] + t + [SEP]$. We represent each token $w_i$ with the representation $h_i^{bert,12}$ obtained from the last layer of BERT as input to our model. Our base model is a 4-layer transformer on this representation, similar to the baseline Transformer(B) in Bai et al. (2020). In the rest of this section, we describe the modifications we make to this transformer to build our three proposed models, *ConsTrans*, *RelConsTrans* and *RelConsTransLG*.

### 2.1 CONSTITUENT-BASED TRANSFORMER (*ConsTrans*)

*ConsTrans* contains a stack of 4 Transformer encoder layers (Vaswani et al., 2017) with Multi-Head Attention (MHA) and a point-wise feed forward sub-layer in each layer. As illustrated in Figure 2a, the encoder stack of *ConsTrans* is grouped into two parts - the lower layers and the upper layers. In all our experiments, we have 2 layers each in both the lower and upper layers. The main difference between a vanilla Transformer network and *ConsTrans* is that the attention scores computed in the MHA layer between a pair of tokens are adjusted based on the probability that the two tokens belong to the same constituent. In the lower layers, attention weights are adjusted such that greater attention weights are assigned to tokens within the same constituent. This adjustment is not imposed at upper layers of the encoder to allow for longer range interactions.

Figure 2b shows a single encoder layer from the lower layers of the encoder stack. The shaded region in the figure, which emphasizes the difference from a vanilla Transformer encoder layer, contains three components: the MHA which provides the vanilla attention scores, the constituent probability scorer, and finally the adjusted MHA scorer that computes the final attention.

**Constituent Probability Scorer** Kim et al. (2020b) found that tokens from the same constituent tend to exhibit similar attention distributions. Hence we propose to determine the probability that a pair of tokens belong to the same constituent by the similarity of their attention distributions. We use the scaled dot-product attention (Vaswani et al., 2017) in the MHA layer to first obtain the attention

distributions of a token:

$$\alpha_{i,j}^{lz} = \frac{\exp F^{lz}(h_i^l, h_j^l)}{\sum_{j'} \exp F^{lz}(h_i^l, h_{j'}^l)}; \quad F^{lz}(h_i^l, h_j^l) = \frac{(W_Q^{lz} h_i^l)(W_K^{lz} h_j^l)^T}{\sqrt{d_k}}, \tag{1}$$

where $W_Q^{lz} \in \mathbb{R}^{d_{model}, d_q}$ and $W_K^{lz} \in \mathbb{R}^{d_{model}, d_k}$ are projection layers that project the query and key to the various attention heads with dimension $d_q$ and $d_k$ respectively. The attention distribution for token $i$ at layer $l$ and attention head $z$ is then defined to be the vector $\alpha_i^{lz} = [\alpha_{ij}^{lz}]_j$. To obtain the attention distribution similarity for two tokens, $i$ and $j$, we concatenate the attention patterns of the pair before passing it through a projection layer:

$$sim_{i,j}^{lz} = \sqrt{\sigma(W_\alpha[\alpha_i^{lz}, \alpha_j^{lz}]) \times \sigma(W_\alpha[\alpha_j^{lz}, \alpha_i^{lz}])} \tag{2}$$

where $sim_{i,j}^{lz}$ refers to the attention distribution similarity score for token $i$ and token $j$ in layer $l$ for attention head $z$, $W_\alpha \in \mathbb{R}^{d,1}$ is a linear projection, $[:,:]$ refers to the concatenation function and $\sigma$ the sigmoid function so that $sim_{i,j}^{lz} \in [0,1]$. We also note that this ensures $sim_{i,j}^{lz} = sim_{j,i}^{lz}$.

We then use the attention distribution similarity scores to compute the probability that a pair of tokens belong to the same constituent. The base probability $c_{i,j}^{'lz}$, that tokens $i$ and $j$ belong to the same constituent, is computed as follows:

$$c_{i,j}^{'lz} = \begin{cases} \prod_{k=0}^{j-i} sim_{i+k,i+k+1}^{lz} & j \leq i \\ \prod_{k=0}^{i-j} sim_{j+k,j+k+1}^{lz} & i > j \end{cases} \tag{3}$$

This formulation considers the probability that tokens spanned by the two tokens $i$ and $j$ form a contiguous constituent. Moreover, since $sim_{i,j}^{lz} \in [0,1]$, the probability that two tokens are in the same constituent would decrease monotonically with the distance between $i$ and $j$. To encourage the induced constituents to be consistent across layers, the final constituent probabilities obtained at the current layer would be the weighted sum of itself and constituent probabilities from the previous layer:

$$c_{i,j}^{lz} = \lambda * c_{i,j}^{'lz} + (1 - \lambda) * c_{i,j}^{'l-1,z}, \tag{4}$$

where $\lambda \in [0,1]$ is a hyper-parameter that we tune.

**Adjusted Attention in the Lower Constituent Layers**  Finally, we adjust the attention scores between a pair of tokens according to the probability that the pair belongs to the same constituent, through a softmax layer:

$$\alpha_{i,j}^{'lz} = \frac{\exp c_{i,j}^{lz} * \alpha_{i,j}^{lz}}{\sum_{j'} \exp c_{i,j}^{lz} * \alpha_{i,j'}^{lz}} \tag{5}$$

where $\alpha_{i,j}^{'lz}$ denotes the adjusted attention score for token $i$ and $j$ for layer $l$ and attention head $z$.

## 2.2 RELATIONAL CONSTITUENT-BASED TRANSFORMER (*RelConsTrans*)

We extend *ConsTrans* with the objective of learning relation embedding between pairs of tokens. For each pair of tokens, the goal is to learn an embedding that represents the syntactic relation between the pair. To generate the relation embedding, we learn a non-linear projection for the concatenation of the representation for the tokens:

$$r_{i,j} = W_{r2} ELU(W_{r1}[h_i, h_j]), \tag{6}$$

where $r_{i,j}$ is the learnt embedding for token $i$ and token $j$, $ELU$ is the exponential linear unit, $h_i$ and $h_j$ are BERT embeddings for token $i$ and $j$ respectively. The learnt embedding would be included in two ways - during attention computation and during information propagation stage.

**Relation-aware Attention Computation**  To perform relation-aware attention computation, we make the following changes to the adjusted scaled dot-product attention formulation in Equation 1 before adjusting the attention scores with constituent probability as in Equation 5:

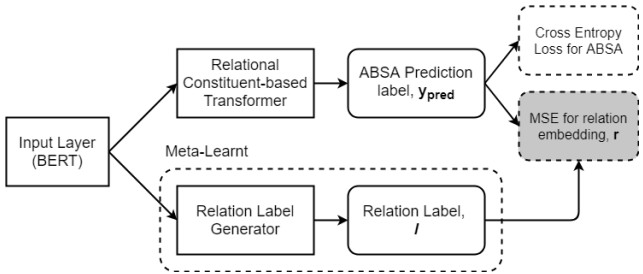

Figure 3: Oveview of Relational Constituent-based Transformer and Relation Label Generator.

$$\alpha_{i,j}^{lz} = \frac{\exp F^{lz}(h_i^l, h_j^l)}{\sum_{j'} \exp F^{lz}(h_i^l, h_{j'}^l)}; \quad F^{lz}(h_i^l, h_j^l) = \frac{(W_Q^{lz} h_i^l)(W_K^{lz} h_j^l + W_{Kr}^l r_{ij})^T}{\sqrt{d_k}} \tag{7}$$

where $W_{Kr}^l \in \mathbb{R}^{d_r, d_k}$ projects the learnt relation embedding $r_{ij}$ and is shared across attention heads. The attention weights would be determined by both textual features, $h_j^l$, and syntactic relation represented by $r_{ij}$.

**Relation-aware information propagation** The attention weights obtained from Equation 7 would then be used to weigh the contribution of other tokens in updating the representation of a token, $h_i^l$ with the following equation:

$$h_i^{l+1} = W_p^l [\sum_j \alpha_{i,j}^{'l1} * (W_V^{lz} h_j^l + W_{Vr}^l r_{ij})]_z \tag{8}$$

where $W_V^{lz} \in \mathbb{R}^{d_{model}, d_v}$ is a projection layer that project the value vector to various attention heads with dimension $d_v$. $W_p^l \in \mathbb{R}^{d_v, d_{model}}$ projects the concatenation of vectors from each attention head to size $d_{model}$ for layer $l$. $W_{Vr}^l \in \mathbb{R}^{d_r, d_v}$ is a projection layer and is shared across attention heads. Therefore, both textual and syntactic and features would be propagated from one token to another.

## 2.3 *RelConsTrans* WITH LABEL GENERATOR (*RelConsTransLG*)

We found that *RelConsTrans* fails to outperform *ConsTrans*, possibly due to a lack of guidance on how the relation embedding should be learned. Previous work (e.g., Bai et al., 2020) used the dependency parser as an auxiliary label generator to improve ABSA. To avoid the need for a dependency parser, we propose to meta learn a relation label generator that would be trained alongside the primary task to generate auxiliary labels optimal for enhancing the performance of ABSA. An overview of the Relational Constituent-based Transformer with the Relation Generator (*RelConsTransLG*) is shown in Figure 3. The relation label generator, trained in a self-supervised manner (Liu et al., 2019), would produce relation labels to guide the learning of the relation embedding.

As syntax information has been shown to be useful for ABSA in previous work (e.g., Bai et al., 2020), we design our relation label generator to encourage the generation of syntax related labels as relation labels with supervision from the ABSA task. Hewitt & Manning (2019) showed that the L2 distance of a linear projection of token embeddings obtained from BERT could recover the parse tree distances between the tokens. Therefore, we learn a linear transformation of the word representation space with the intention of learning syntactic relatedness. The learned syntactic relatedness would then be used as the ground truth for the L2 norm of the relation embedding. Different from Hewitt & Manning (2019), we do not use ground truth labels to train the relation label generator. Instead, we learn this linear projection in a meta learning manner.

For a pair of tokens $i$ and $j$, we learn a linear projection for the BERT representation:

$$l_{ij} = W_1(h_i^{bert,n} - h_j^{bert,n}) + b_1, \tag{9}$$

where $l_{ij}$ is a scalar relation label for token $i$ and $j$, $W_1$ and $b_1$ are the weights and bias of the linear transformation layer. $h_i^{bert,n}$ represents the embedding of token $i$ from the $n^{th}$ layer of the BERT

| Data Set | Positive | | | Neutral | | | Negative | | |
|---|---|---|---|---|---|---|---|---|---|
| | Train | Dev | Test | Train | Dev | Test | Train | Dev | Test |
| **Restaurant** | 2164 | - | 728 | 637 | - | 196 | 807 | - | 196 |
| **Laptop** | 994 | - | 341 | 464 | - | 169 | 870 | - | 128 |
| **Twitter14** | 1561 | - | 173 | 3127 | - | 346 | 1560 | - | 173 |
| **Twitter15** | 928 | 303 | 317 | 1883 | 670 | 607 | 368 | 149 | 113 |
| **Twitter17** | 1508 | 515 | 493 | 1638 | 517 | 573 | 416 | 144 | 168 |
| **Twitter14 (AS)** | 1538 | - | 190 | 3300 | - | 173 | 1445 | - | 288 |

Table 1: Statistics of the 5 benchmark data sets. **TS** refers to splitting the data by aspect.

model. As different layers of BERT appear to represent different types of information as shown by Tenney et al. (2019), we could recover different information by selecting different BERT layers (different $n$). In our experiments, we fixed the value of $n$ to 6 for all data sets.

**Meta Training Relation Label Generator**   To guide the training of the embedding, we minimize the mean square error (MSE) of the generated label and the L2 norm of the relation embedding:

$$MSE(l, r) = \sum_{i,j}(\|r_{ij}\|_2 - l_{ij})^2. \tag{10}$$

Drawing inspiration from recent work by Liu et al. (2019), we train our label generator using the loss from ABSA with the goal of generating relation labels $l_{ij}$ to directly optimize for the performance of the main task.

Let $\theta_{main}$ be the parameters of our main model, *RelConsTrans*. To update the parameters of $\theta_{main}$, we aim to minimize a multi-task loss – cross-entropy loss, $L$ from the ABSA prediction task and the MSE loss described in Equation 10:

$$\underset{\theta_{main}}{\arg\min}(L(\hat{y}, y) + MSE(l, r)). \tag{11}$$

Let $\theta_{main}^+$ be the weights of the *RelConsTrans* after one gradient update step of gradient descent:

$$\theta_{main}^+ = \theta_{main} - \alpha_{main}\nabla_{\theta_{main}}\underset{\theta_{main}}{\arg\min}(L(\hat{y}, y) + MSE(l, r)), \tag{12}$$

where $\alpha_{main}$ is the learning rate to train the *RelConsTrans*. Note that the MSE from the relation embedding would not be used to train the relation label generator. Therefore, the parameters of the relation label generator, $\theta_{aux}$ should be updated by solely the loss from ABSA:

$$\underset{\theta_{aux}}{\arg\min}(L(\hat{y}, y)), \tag{13}$$

To update the weights of the generator, a second order derivative is computed. While this formulation was inspired by Liu et al. (2019), the second-order derivative trick used in our model was also used in a number of other meta-learning frameworks such as Finn et al. (2017).

We train the two models in tandem, over a few iterations. We found it useful to train $\theta_{main}$ and $\theta_{aux}$ with separate training sets. For each data set, we took a subset of the cases which contain two or more aspects (meta-train set) in the same sentence for training $\theta_{aux}$. This subset is removed from the main training set (train set) used to train the main *RelConsTrans*. More details of the meta-train set would be provided in the appendix A.1.

## 3   RESULTS AND ANALYSIS

We conducted experiments on 5 benchmark data sets - restaurant reviews, laptop reviews from SemEval 2014 (Pontiki et al., 2014), ACL14 Twitter14 data set (Dong et al., 2014) and Twitter15 and Twitter17 from a multi-modal ABSA data set by (Yu & Jiang, 2019). For analysis, we ran additional experiments on a split of the Twitter14 data set by aspect. We summarize the statistics of the data in Table 1. For data sets with development sets, we perform model selection on the development sets.

For Restaurant, Laptop and Twitter14, we compare against published results from BERT-PT (Xu et al., 2019), BERT-SPC (Song et al., 2019), AEN-BERT (Song et al., 2019), SDGCN-BERT (Zhao

| Data Set Model | Restaurant Acc | F1 | Laptop Acc | F1 | Twitter14 Acc | F1 |
|---|---|---|---|---|---|---|
| **BERT-PT** | 85.0 | 77.0 | 78.1 | 75.1 | - | - |
| **BERT-SPC** | 84.5 | 77.0 | 79.0 | 75.0 | 73.6 | 72.1 |
| **AEN-BERT** | 83.1 | 73.8 | 79.9 | 76.3 | 74.7 | 73.1 |
| **SDGCN-BERT** | 83.6 | 76.5 | 81.4 | 78.3 | - | - |
| **Transformer(B)** | 84.9 | 77.9 | 79.3 | 76.1 | - | - |
| **RGAT-Bai\*** | 86.6 | 80.5 | 81.3 | **78.6** | 75.8 | 74.7 |
| **RGAT-Wang\*** | 86.6 | **81.4** | 78.2 | 74.1 | 76.2 | 74.9 |
| **RGAT-Wang (re-run)\*** | 85.7 | 79.1 | 79.0 | 75.6 | 73.6 | 73.1 |
| **DGEDT-BERT\*** | 86.3 | 80.0 | 79.8 | 75.6 | 77.9 | 75.4 |
| **LCFS-ASC-CDW\*** | 86.7 | 80.3 | 80.5 | 77.1 | - | - |
| *ConsTrans* | 85.8 | 80.8 | 80.6 | 77.2 | 76.6 | 75.0 |
| *RelConsTrans* | 85.4 | 79.3 | 80.1 | 76.4 | 75.9 | 74.7 |
| *RelConsTransLG* | 86.7 | **81.4** | 81.0 | 78.1 | 76.9 | **75.5** |

| Data Set Model | Twitter15 Acc | F1 | Twitter17 Acc | F1 |
|---|---|---|---|---|
| **AE-LSTM** | 70.3 | 63.4 | 61.7 | 58.0 |
| **MemNet** | 70.1 | 61.8 | 64.2 | 60.9 |
| **RAM** | 70.7 | 63.1 | 64.4 | 61.0 |
| **MGAN** | 71.2 | 64.2 | 64.8 | 61.5 |
| **BERT** | 74.2 | 68.9 | 68.2 | 65.2 |
| **BERT+BL** | 74.3 | 70.0 | 68.9 | 66.1 |
| **TomBERT\*** | 77.2 | 71.8 | 70.5 | 68.0 |
| *ConsTrans* | 76.5 | 72.5 | 69.3 | 68.2 |
| *RelConsTrans* | 76.9 | 71.6 | 69.0 | 67.7 |
| *RelConsTransLG* | 76.8 | **73.3** | 69.8 | **68.5** |

Table 2: Accuracy and F-score (F1) for 5 data sets: In the left (resp. right) table, systems marked * are those that used dependency parses (resp. multi-modal information). The best Macro F1 for each data set is in **bold**. For significance tests, we compare against RGAT-Wang(re-run), TomBERT and BERT+BL. Our results are significant against RGAT-Wang(re-run) and BERT+BL. Twitter17 was significant against TomBERT (which used image data in addition to text data) but not Twitter15.

et al., 2020), Transformer(B) (Bai et al., 2020), RGAT-Bai (Bai et al., 2020), RGAT-Wang (Wang et al., 2020), DGEDT-BERT (Tang et al., 2020) and LCFS-ASC-CDW (Phan & Ogunbona, 2020). The Transformer(B) is a baseline model used by Bai et al. (2020), and is the baseline vanilla Transformer on which *ConsTrans* is built upon. For Twitter 15 and Twitter17, we compare against published results in (Yu & Jiang, 2019): MemNet (Tang et al., 2016), RAM (Chen et al., 2017), MGAN (Fan et al., 2018), BERT, BERT+BL (Yu & Jiang, 2019) and TomBERT (Yu & Jiang, 2019).

For the Restaurant and Twitter14 data sets, we outperform all previous work that did not require dependency parsers by competitive margins (4.8 F-score for Restaurant and 2.4 F-score for Twitter14). Our results on the Laptop data (78.1) is also close to the state-of-the-art results (78.3) achieved by SDGCN-BERT. Furthermore, comparing results with models that require a dependency parser, we also outperform a number of models while closely matching the results of others. For Twitter15 and Twitter17, we see in Table 2 that our best model outperforms previous work that uses only textual content by a margin (3.3 F-score for Twitter15 and 2.4 F-Score for Twitter17). Our model also outperforms TomBERT, the multi-modal models for Twitter15 and Twitter17.

To conduct statistical significance tests, we attempt to reproduce the results published for RGAT-Wang, TomBERT and BERT+BL. For RGAT-Wang, we could not reproduce their published results (with their recommended settings), and hence we can only conduct the test against the results we obtain with their open source code, shown as RGAT-Wang(re-run) in Table 2. We run the randomization test (Yeh, 2000) with 100,000 shuffles. We found that *RelConsTransLG* outperforms RGAT-Wang(re-run) and BERT+BL significantly ($p < 0.15$). *RelConsTransLG* significantly outperforms TomBERT (which has additional access to image data) for Twitter17, but not Twitter15.

When comparing our proposed *ConsTrans* model to a vanilla Transformer, we observe that *ConsTrans* outperforms the vanilla Transformer model for both the Restaurant and Laptop data sets. This suggests that it is indeed useful to induce constituents for ABSA. Lastly, comparing *ConsTrans* and *RelConsTransLG*, we observe that *RelConsTransLG* consistently outperforms *ConsTrans* for all the data sets. This suggests that our meta-learnt label generator is able to generate useful auxiliary labels for *ConsTrans* for the ABSA task.

### 3.1 ANALYSIS

In this section, we provide findings from ablation studies and analysis of our proposed models.

**Grammar Induction** We derive constituent trees with the constituent probabilities to verify if the derived trees resemble ground truth constituent trees. In Figure 4, we show an example where our derived constituent tree shows a similar structure to the ground truth constituency tree. Notably, we are able to accurately recall the aspect term, *"jessica alba"* as a constituent. The algorithm to derive constituent trees and more examples are provided in Appendix A.4 and A.6 respectively.

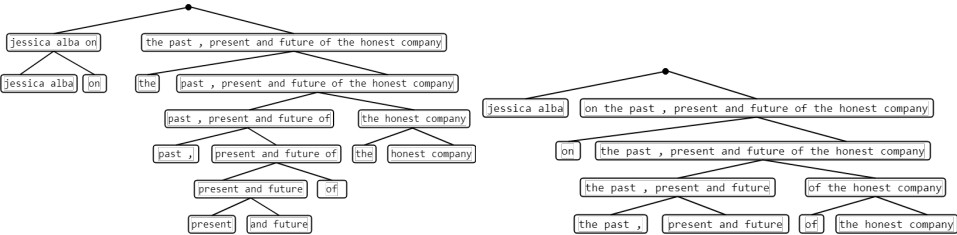

Figure 4: Example of derived constituent Tree by *ConsTrans* (Left) and constituent Tree from Berkeley Neural Parser (Kitaev & Klein, 2018) (Right).

We postulate that the ability of *ConsTrans* to group aspect terms into noun phrases should improve its ABSA accuracy. To study our hypothesis, we looked at *ConsTrans*'s ability to recall the entire aspect term as a noun phrase. For simplicity, we only look at records where aspect terms were not broken down into sub-word tokens. For Twitter17, we found that *ConsTrans* achieves 67.8 recall rate for correctly predicted instances and 62.2 for incorrect instances. The Pearson correlation coefficient between prediction accuracy and recall rate was significant (with $p < 0.2$), indicating the usefulness of being able to induce good constituents for ABSA.

**Generalibility of *RelConsTransLG*** The key argument by Liu et al. (2019) for designing an additional label generator is to increase the generalizability of the main model. To test the generalizability of *RelConsTransLG*, we create a more challenging version of Twitter14 by splitting the data such that the train and test set comprises of different aspect terms. The statistics of the data after splitting by aspect (denoted by AS) is shown in Table 1. We further split the train set by aspect terms to create a meta-train set to train the label generator in *RelConsTransLG*. Therefore, the relation label generator is trained to generate relation labels that enhance the performance of data with foreign aspect terms. In this AS setting, *RelConsTransLG* achieves a F-score of 64.3 while *ConsTrans* achieved a F-score of 62.8. Our designed framework mimics the actual train and test setting and is therefore able to increase the generalizabillity of *RelConsTransLG*.

**Different layers of BERT as input** Tenney et al. (2019) found that different BERT layers encapsulate different information useful for various NLP tasks. Therefore, we experimented with using all 12 layers of BERT as input to the label generator to study the impact on the F-score. The graph for the F-score against BERT layer ($n$) is provided in Appendix A.3. Using representation from the $6^{th}$ layer of BERT yields the best results for the restaurant data set and we are able to consistently outperform models that do not use dependency parses for all value of $n$ chosen. Furthermore, this is an indication that syntactic information is indeed useful for ABSA since lower layers of BERT were found to encapsulate syntactic information.

**Interpreting learnt relation labels** Our relation label generator is designed to encourage the generation of syntax related labels. To verify the hypothesis that generated relation label is related to syntax, we reconstruct the dependency parses using the learnt relation embedding. Interestingly, while Dozat & Manning (2017) have found that the L2 norm of relation embedding resembles syntactic distance (i.e., a lower norm means stronger dependency), we found that our learned relation embedding exhibits an opposite phenomenon: a higher L2 norm indicates a stronger dependency. We hypothesize that relation embedding with higher L2 norm would influence attention weights to a greater extent. Therefore, the L2 norm of our learnt relation embedding would represent syntactic relatedness rather than syntactic distance. We then construct parse trees by linking tokens with highest L2 norm of their relation embedding as detailed in Appendix A.5.

Manual inspection of these records suggest that while a full parse tree was not induced, we are able to recover most of adjective-noun relations. As seen in Figure 5, we are able to retrieve the relation of (*"pleasant"*, *"staff"*) and (*"friendly"*, *"staff"*). This is expected since understanding adjective-noun relations would be most important to ABSA compared to other types of relations. Therefore, training *RelConsTransLG* with supervision from solely ABSA would yield this behaviour.

Furthermore, to look at the ability of *RelConsTransLG* to link relevant adjective terms, we engaged two annotators to annotate the adjective terms relevant to each aspect term for the test set for the

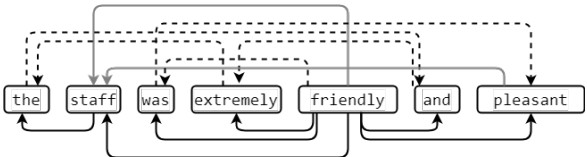

Figure 5: Examples of induced dependency parses Tree by *RelConsTransLG* (Top arrows) and ground truth dependency parses (bottom arrows) from StanfordNLP https://corenlp.run/. Arrows in grey are for opinion terms accurately linked to the aspect term "staff".

Restaurant data. The annotators reconciled their differing opinions and gave each record a final label. Records with no clear opinion terms was given a "None" label. There were 1,120 records annotated and 811 had annotated adjective terms. We rank the relatedness of tokens with the aspect term by the L2 norm of the relation embedding and compare it with ground truth ranks. Ground truth ranks were obtained by ranking tokens with their syntactic distance obtained from StanfordNLP dependency parser (Chen & Manning, 2014) with tied rank taken into account. For records where the adjective term was more than 1 syntactic distance away, we obtain an equal or smaller rank than the syntactic distance in 63.5% of the cases. Compared against position offset ranks, we obtain an equal or small rank than the number of position offsets in 65.0% of the cases.

## 4 RELATED WORK

Sentiment analysis (Pang & Lee, 2008) is a well studied natural language processing problem. Early works applied sentiment analysis to product reviews as a text classification problem. However, a review or social media post could express different sentiments to different aspects, and the task of aspect-based sentiment analysis aims at a finer classification of sentiment towards specific aspects (Dong et al., 2014) or aspects (Pontiki et al., 2016).

Recent work on ABSA has shown that the use of dependency parses for ABSA helps to improve performance (Bai et al., 2020; Huang & Carley, 2019; Sun et al., 2019; Wang et al., 2020). However, supervised dependency parsers require a substantial amount of annotated data, and might perform badly for out-of-domain (e.g., social media) or low-resource languages. On the other hand, it has been shown that contextual embeddings such as BERT (Devlin et al., 2018) contain significant information that could be useful to parsers (Clark et al., 2019; Kim et al., 2020a). Previous work such as Hewitt & Manning (2019) have shown that a linear projection is sufficient to recover syntactic information from BERT embedding. In this paper, we show that we can achieve similar ABSA performance without supervised parsers, by leveraging on BERT which was trained with raw data.

Previous work on unsupervised grammar induction such as Shen et al. (2019); Kim et al. (2019) aims to induce grammar from raw data. Our primary objective is not to induce grammar, but to encourage the model to learn to perform the ABSA task by learning the causal edge dependencies between constituents. We show that our approach is able to achieve results that rivals those obtained by models that have access to supervised dependency parsers.

In this work, we applied meta auxiliary learning (Liu et al., 2019) which learns to generate auxiliary labels, supervised by the primary task. While Liu et al. (2019) failed to interpret the auxiliary labels for the computer vision tasks they worked on, we showed that in our case, the induced auxiliary labels can be interpreted as syntactic relatedness to a certain extent.

## 5 CONCLUSION

In this paper, we apply a meta auxiliary learning approach to the ABSA task, and we show that the induced relations between phrases are interpretable and supports the primary task of sentiment analysis. We show that learning the auxiliary labels improve results over our baselines on all five data sets. Without using dependency parsers, our approach performs competitively compared to previous work that used dependency parses as input.

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

# A  APPENDIX

## A.1  META-SPLITTING DATA SET

To generate the meta-train set, we first obtain a subset of samples with multiple aspects of differing sentiment from the full train set. 80% of this subset would then be used as the meta-train set. For Twitter2014 (AS), we sampled this meta-train set by aspects. The intention is to train the Relation Label Generator to generate labels that can encourage the model to accurately link relevant opinion terms to the aspect term through challenging samples. The statistics of the meta-train set is provided in Table 3.

| Data set | Positive | Neutral | Negative | Total |
|---|---|---|---|---|
| **Restaurant** | 195 | 163 | 162 | 520 |
| **Laptop** | 85 | 98 | 61 | 244 |
| **Twitter2014** | 265 | 567 | 301 | 1133 |
| **Twitter2015** | 123 | 161 | 56 | 340 |
| **Twitter2017** | 320 | 470 | 111 | 901 |
| **Twitter2014 (AS)** | 207 | 253 | 254 | 714 |

Table 3: Statistics of the meta-train sets. **AS** refers to splitting the data by aspect.

## A.2  IMPLEMENTATION DETAILS

In all experiments, we used the ADAM optimizer (Kingma & Ba, 2015) with a learning rate of $10^{-5}$ to optimize the models (*ConsTrans*, *RelConsTrans*, *RelConsTransLG* and the relation label generator). The BERT model was fine-tuned together with the main model built on top of it. We used 6 multi-head attention heads and the size of the hidden layer was set to be 384. We stacked 4 encoder layers. Lower layers are defined to be the first 2 layers in the encoder stack while the last 2 layers are higher layers. Dropout was applied to all input embeddings. The L2 penalisation term for the model's parameters was set to be 1e-5. The size of the learnt embedding for each relation label was set to be 786 and a batch size of 4 was used to train all the models.

## A.3  DIFFERENT LAYERS OF BERT

The graph for Macro F-score against BERT layer ($n$) is shown in Figure 6. Using representation from the $6^{th}$ layer of BERT yields the best results for the restaurant data set and we used the $6^{th}$ layer of BERT as input to the relation label generator for all the other data sets.

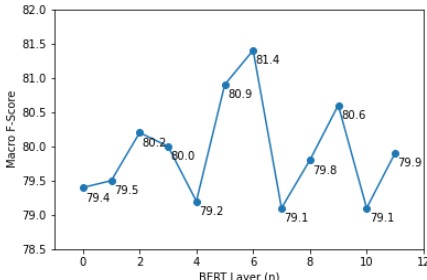

Figure 6: Macro F-score on Restaurant data set with different BERT layer as input to LG.

## A.4  ALGORITHM TO DERIVE CONSTITUENT TREES

To derive constituent trees, we iteratively split the input constituents into 2 parts till each constituent is only made up of only 1 token. The breaking point is defined to be the point where the probability of being in the same constituent for token $i$ and $i+1$ is the lowest. We use the constituent probability from the second layer of our encoder stack to find the breaking point.

---

**Algorithm 1** Unsupervised constituent tree derivation

---

1:  $m \leftarrow$ Constituent layer chosen
2:  $x \leftarrow$ Input tokens
3:  $p \leftarrow$ Constituent probability from layer m for current constituent
4:  $stack = [x]$
5:  $constituents = []$
6:
7:  **procedure** GETCONSTTREE$(x, p)$
8:     **while** len(stack) != 0 **do**
9:        $current\_const \leftarrow$ stack[0]
10:       $p_0 \leftarrow$ constituent probabilities for $current\_const$
11:       b = **argmin**$(b_0)$                                          ▷ Find the breaking point
12:       $left\_const$ = x[:b]
13:       $right\_const$ = x[b:]
14:       **if** len(left_const) $> 1$ **then**
15:           **add** $left\_const$ to $constituents$
16:       **end if**
17:       **if** len(right_const) $> 1$ **then**
18:           **add** $right\_const$ to $constituents$
19:       **end if**
20:       **add** $left\_const$ and $right\_const$ to $constituents$
21:     **end while**
22:     Return $constituents$
23: **end procedure**

---

### A.5   ALGORITHM TO INDUCE DEPENDENCY PARSES

To recover parse trees, we iteratively linked tokens to another token with the highest L2 norm for the learnt relation embedding. We do not allow tokens to be linked multiple times.

---

**Algorithm 2** Unsupervised dependency parse tree induction

---

1:  **procedure** GETDEPNTREE$(x, p)$
2:     $depen\_labels = []$
3:     $relation\_norm \leftarrow$ The L2 norm of learnt embedding
4:     **for** $j \leftarrow 1$ to $length_x$ **do**
5:       $current\_max$ = **max**$(relation\_norm)$
6:       $i, j$ = **position**$(current\_max)$;                           ▷ Get the position of max L2 norm
7:       **add** $(i, j)$ to $depen\_labels$
8:       $relation\_norm$[:, j] $= -\infty$;                        ▷ Do not link to token j
9:       $relation\_norm$[j, i] $= -\infty$;                        ▷ Do not allow loops
10:     **end for**
11:     Return $depen\_labels$
12: **end procedure**

---

### A.6   EXAMPLES OF DERIVED CONSTITUENT TREES

We provide more examples of constituent trees derived from *ConsTrans*. In general, while we were not able to fully replicate ground truth constituent trees, we noticed that the model was able to recall noun phrases reasonably.

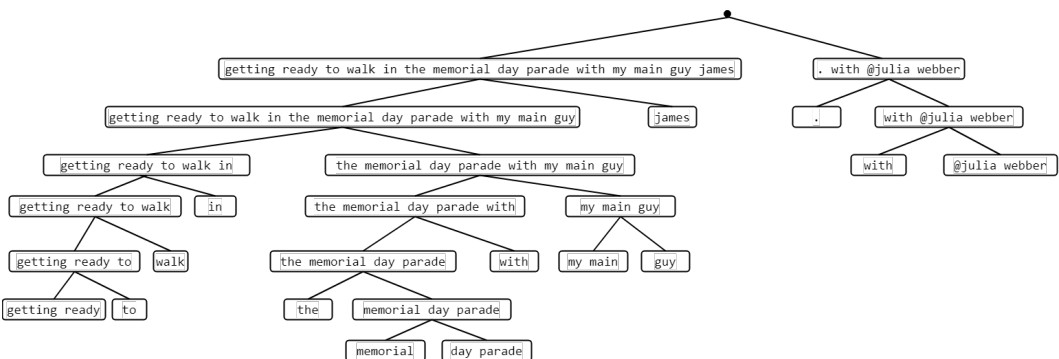

Figure 7: Example of derived constituent Tree by *ConsTrans* with aspect term "@Jullia webber".

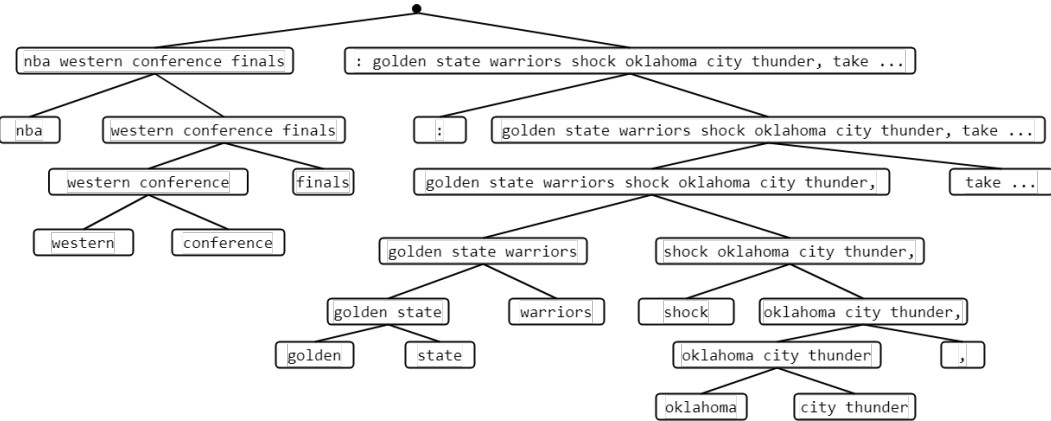

Figure 8: Example of derived constituent Tree by *ConsTrans* with aspect term "@golden state warriors".

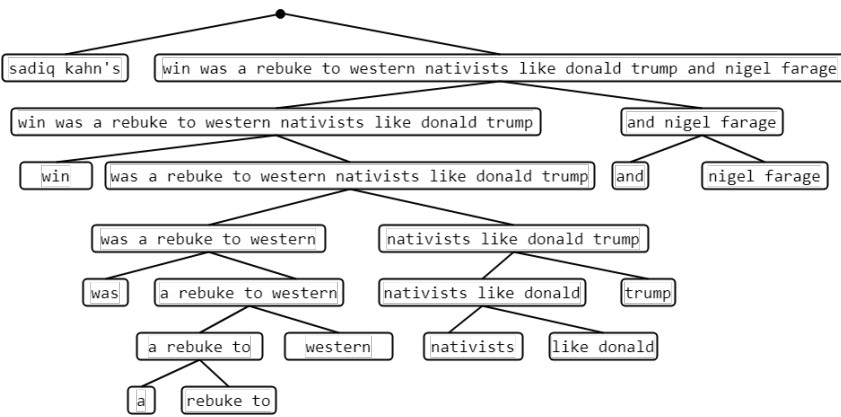

Figure 9: Example of derived constituent Tree by *ConsTrans* with aspect term "sadiq Kahn's".

## A.7    EXAMPLES OF INDUCED DEPENDENCY PARSE TREES

We provide more examples of induced dependency parses from *RelConsTransLG*. While our induced parse trees were not identical to ground truth parse trees, we were able to link adjective terms to noun phrases reasonably.

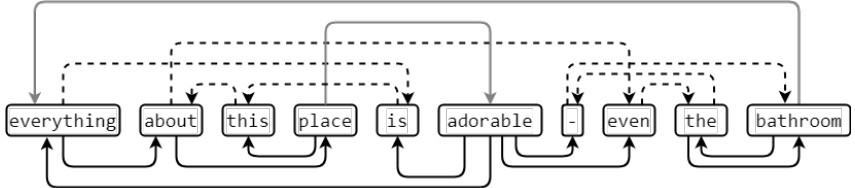

Figure 10: Examples of induced dependency parses Tree by *RelConsTransLG* (Top arrows) and ground truth dependency parses (bottom arrows) from StanfordNLP https://corenlp.run/. Arrows in grey are for opinion terms accurately linked to the aspect term "place".

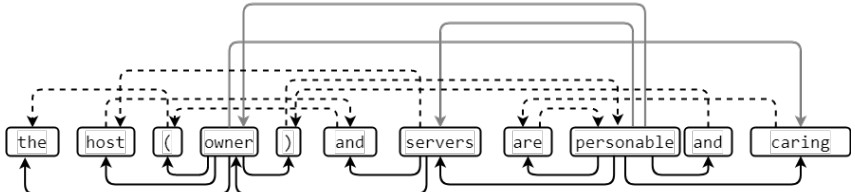

Figure 11: Examples of induced dependency parses Tree by *RelConsTransLG* (Top arrows) and ground truth dependency parses (bottom arrows) from StanfordNLP https://corenlp.run/. Arrows in grey are for opinion terms accurately linked to the aspect term "owner".

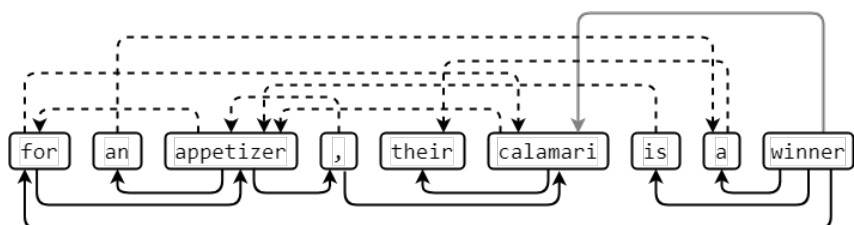

Figure 12: Examples of induced dependency parses Tree by *RelConsTransLG* (Top arrows) and ground truth dependency parses (bottom arrows) from StanfordNLP https://corenlp.run/. Arrows in grey are for opinion terms accurately linked to the aspect term "calamari".

