# OpenReview forum: "Meta Auxiliary Labels with Constituent-based Transformer for Aspect-based Sentiment Analysis"
_ICLR.cc/2021/Conference — Reject_

### Official Review · AnonReviewer3 · 2020-10-23
**This paper studies the model design of removing explicit syntax from sentiment analysis task, but it also has some limitations.**

**Rating:** 4
**Confidence:** 4

**Review:**

According to two phenomena in Transformer's pre-trained language model:
1. Tokens with similar self-attention distribution tend to be distributed in the same constituent;
2. The L2 distance between token representations reveals the degree of syntactic dependency;

an aspect-based sentimental analysis system is designed which does not rely on explicit syntactic information. It is not a surprising point to combine the so-called syntactic information learning with sentential analysis. Although the model has achieved comparable results with systems using explicit syntax, the meta-learning is somewhat overclaimed.

There are some defects in this work:
1. Over dependence on the pre-trained language model: Although the author claims that explicit syntactic information is not used, the system relies on the syntactic information contained in the pre-trained language models such as BERT to promote the training of ConsTrans, RelConsTrans, and RelConsTransLG. I doubt whether the training of the model can converge without the pre-trained language model only relying on the randomly initialized transformer and still achieve comparable accuracy with systems with explicit syntax? What is the performance of using a weaker pre-trained language model like Distibert? How about the performance in stronger pre-trained models like RoBERTa?
2. Model size and inference performance: the extra transformer layers added makes the result not directly comparable with that of BERT. Therefore, the author needs to report the effect of the BERT baseline under the same parameter scale. In addition, to what extent will the introduction of the constituent probability scorer and relation label generator affect the inference performance of the model? For this simple task of sentiment analysis, do we need such a complex model design and training method design after the already huge pre-trained language model such as BERT?
3. Inappropriate description:
(1) In RelConsTrans, the proposed model encodes the syntactic relation as a feature for constituent scoring based on the role of syntactic relation for sentiment analysis. But according to Eq. 6 and 7, the so-called relation (that could not be determined) act as a bias in scoring function, I think it just provides a more feature source (dependency syntax) for scoring.
(2) In RelConsTransLG, the r_{i,j} and l_{i,j} is similar. I think the only conclusion that can be drawn is that the model is further enhanced with the features of dependency syntax recently, which has little relationship with meta-learning.

Generally speaking, this work can be regarded as integrating the implicit constituent and dependency syntactic features from the pre-trained language models into the training in the sentient analysis, so that explicit syntax is no longer needed. Appendix A.6 and A.7 also confirm this point. Although it has a positive effect on the need to remove explicit syntax, its reliance on pre-trained language models limits the extensibility of this approach and is likely to be ineffective on traditional unpretrained LSTM, CNN, and Transformer models. In addition, since the writing is pretended to make the model close to a concept of meta-learning, it ignores the important points that should be clarified.

---

> ### Author Response · Authors · 2020-11-15
> **Response to AnonReviewer3**
>
> Thank you for taking time to review the paper and for the constructive feedback provided. We address your questions and comments below:\
> \
> Question 1:\
> a.	Over dependence on the pre-trained language model: Although it was claimed that explicit syntactic information is not used, syntactic information contained in the pre-trained language models such as BERT was used to promote the training of ConsTrans, RelConsTrans, and RelConsTransLG. \
> b.	What is the performance of using a weaker pre-trained language model like Distibert? How about the performance in stronger pre-trained models like RoBERTa?\
> \
> Response to Q1a:\
> We agree that we are exploiting syntactic information in BERT to replace dependency parsers. We have shown that this is useful for Twitter, where we outperformed SOTA results that use dependency parsers on ABSA. This is also useful for resource-poor languages: it is easier to build BERT for a resource poor language than to build a dependency parser, since SOTA dependency parsers require annotated dependency data sets, while BERT only requires raw unlabelled data.
>
> Response to Q1b:\
> We have explored using a stronger model (XLNET) and there was a performance gain of approximately 2% F-score across all the datasets. However, for fair comparison against previous SOTA on the ABSA task (e.g., [1] and 2]), we have only reported results for BERT. \
> We would provide results for weaker pre-trained language models like DistilBERT in the revised version.
>
> Question 2:\
> a.	Model size and inference performance: the extra transformer layers added makes the result not directly comparable with that of BERT. Therefore, effect of the BERT baseline under the same parameter scale needs to be reported.\
> b.	To what extent will the introduction of the constituent probability scorer and relation label generator affect the inference performance of the model? \
> c.	For this simple task of sentiment analysis, is such a complex model design and training method design needed after the already huge pre-trained language model such as BERT?
>
> Response to Q2a:\
> Our baseline model is not a simple BERT classifier. Our baseline models are equally complex models such as RGAT-Wang [1] and LCFS-ASC-CDW [2], which built transformer layers on top of BERT as we have done. Moreover, these baselines require additional inputs such as dependency parses, or multi-modal inputs (images as well as text).
>
> Response to Q2b:\
> We have shown that we consistently outperform a vanilla transformer with the same number of layers on 2 datasets (Restaurant and Laptop) with the addition of the constituent probability scorer (ConsTrans). The label generator is also shown to consistently outperform the model trained without it.
>
> Response to Q2c:\
> Our task is Aspect based Sentiment Analysis (ABSA) and not general Sentiment Analysis (SA) The ABSA task is a rich and difficult NLP task: for example, to classify “I will migrate to Europe if Trump gets re-elected” as negative towards “Trump” requires contextual information, and the sentiment changes from negative to positive if we change “Europe” to “the US”. There is a lot of literature on ABSA and we have shown that we have outperformed all previous work including two recent papers published in ACL 2020 [1,2].
>
> Question 3:\
> The only conclusion that can be drawn is that the model is further enhanced with the features of dependency syntax recently, which has little relationship with meta-learning.
>
> Response to question 3:\
> We are applying the meta-auxiliary learning approach from [4] to learn auxiliary relation labels. This is different from the vanilla meta-learning from multiple supervised tasks. In [4], they used the term meta-learning as they argued: “leveraging the performance of the multi-task network to train the label-generation network can be considered as a form of meta learning”.
>
> Response to comments:\
> We have attempted to address the comments in our responses for question 1 and 3.
>
> [1] Relational graph attention network for aspect-based sentiment analysis. Kai Wang, Weizhou Shen, Yunyi Yang, Xiaojun Quan, and Rui Wang. ACL 2020.\
> [2] Modelling context and syntactical features for aspect based sentiment analysis. Minh Hieu Phan and Philip O. Ogunbona. ACL 2020.\
> [3] Adapting BERT for target-oriented multimodal sentiment classification. Jianfei Yu and Jing Jiang. IJCAI 2019.\
> [4] Self-Supervised Generalisation with Meta Auxiliary Learning. Shikun Liu, Andrew J. Davison, Edward Johns. Neurips 2020.

---

### Official Review · AnonReviewer1 · 2020-10-26
**Interesting modification on transformer, but limited applications and improvements**

**Rating:** 3
**Confidence:** 5

**Review:**

This paper presents a constituent-based transformer model with auxiliary relation embeddings and labels to enhance the performance of aspect-based sentiment analysis. The constituent-based transformer modifies the original transformer by re-weighting the attention weights with constituent-based similarities. The auxiliary relation predictions helps to differentiate the relations that are useful for sentiment prediction.

In general, this paper is readable and has a clear motivation. The strengths include:
1. The idea of modifying the basic transformer to a constituent-based transformer by incorporating constituent similarities without any supervision is interesting.
2. The authors propose three model variations progressively, namely ConsTrans, RelConsTrans and RelConsTransLG to clearly demonstrate the motivation and effect of each variation.

However, the paper still lack the following aspects:
1. Some parts of the description is unclear with details missing. From (9), the generated $l_{ij}$ is a continuous value, how do you make it a relation label? Why do you take the L2 norm on $r_{ij}$ in (10)? What is the intuition of such computation to obtain the MSE? What is $L(\hat{y},y)$ in (11) and how to compute it? If $y$ is the sentiment label of the aspects, how it could be used to update the label generator $l_{ij}$? What is the intuition of training $\theta_{main}$ and $\theta_{aux}$ using separate datasets? Could you show what is the difference in terms of the performance?
2. Given the complexity of the model, the improvement compared to the baseline models are relatively trivial. And it is somewhat insufficient to limit the application only to aspect-based sentiment analysis. The contribution is thus limited.
3. How do you sample the meta-train set? Is the result over one meta-train set or averaged over different meta-train set?
4. Minor points: caption is missing for figure 2. The first row in (3) should be $i\leq j$.

---

> ### Author Response · Authors · 2020-11-15
> **Response to AnonReviewer1**
>
> Thank you for taking time to review the paper and for the constructive feedback provided. We address your questions and comments below:\
> \
> Question 1: \
> a.	The generated $l_{ij}$ is a continuous value, how can it be made into a relation label? \
> b.	Why was the L2 norm on $r_{ij}$ in (10) taken? \
> c.	What is $L(\hat{y} , y)$ in (11) and how is it computed? \
> d.	How could it [y (sentiment label of the aspects)] be used to update the label generator  $l_{ij}$?\
> e.	What is the intuition of training $\theta_{main}$ and $\theta_{aux}$ using separate datasets? Could the difference in terms of the performance be shown?\
> \
> Response to Q1a & Q1b:\
> Your observation that  $l_{ij}$ is a continuous value is correct, and we apologize for not clarifying this point in the paper: while relation label is conventionally a categorical value (in dependency parses), we have chosen to train our model to learn scalar relational labels.\
> We took inspiration from [1] where the authors observed that the L2 norm of the linear projection of token embeddings obtained from BERT could recover the parse tree distances between the tokens. Hence, we train the relation scalar label by supervising it with the L2 norm of the relational embedding with the MSE loss.
>
> Response to Q1c:\
> $L(\hat{y} , y)$ in (11) is the cross entropy loss between the ABSA ground truth and the model prediction for ABSA. \
> \
> Response to Q1d:\
> A sketch of our learning framework is as follows: \
> (1) We first train an ABSA predictor with the cross entropy loss, and \
> (2) we then train the label generator using a separate training set. The label generator is optimized such that if the ABSA predictor is trained with the labels produced by the label generator, the loss for the ABSA task is minimized. \
> The model parameters of the label generator $\theta_{aux}$ is frozen in optimizing (11). The label generator is only trained in optimizing $\theta_{aux}$ in Eqn (13).
>
> Response to Q1e:\
> The label generator is trained on a separate set to encourage labels generated to be optimal for a dataset unseen by the ABSA predictor during training. This would encourage the label generator to generate labels useful for generalization since the train setting is similar to the test setting. \
> We sampled this unseen set by selecting records that fulfil the following conditions:
> 1.	Contains more than 1 aspect term in the sentence and
> 2.	The aspect terms have a different sentiment labels
> We argue that these are more “difficult” examples, and having a relation label that represents the relationship between tokens would be exceptionally useful.
>
> Question 2: \
> Given the complexity of the model, the improvement compared to the baseline models are relatively trivial. And it is somewhat insufficient to limit the application only to aspect-based sentiment analysis. The contribution is thus limited.
>
> Response to question 2:\
> Our baseline models are equally complex models such as RGAT-Wang (Wang et al., 2020) and LCFS-ASC-CDW (Phan & Ogunbona, 2020), which built transformer layers on top of BERT as we have done. Moreover, these baselines require additional inputs such as dependency parses or multi-modal inputs (images as well as text).
>
> Question 3:\
> How was the meta-train set sampled? Is the result over one meta-train set or averaged over different meta-train set?
>
> Response to question 3:\
> We have attempted to answer this question in our response to Q1d and Q1e.
>
> Response to comments:\
> We will revise the paper to address the comment of missing caption and error in equation 3.
>
> [1] A Structural Probe for Finding Syntax in Word Representations. John Hewitt, Christopher D. Manning. NAACL 2019

---

### Official Review · AnonReviewer2 · 2020-10-28
**Overly complex model, conceptually unclear, and not well evaluated**

**Rating:** 2
**Confidence:** 4

**Review:**

This paper presents a Transformer-based model for aspect-based sentiment analysis, intended to support the unsupervised induction of constituents within the Transformer forward pass. Their evaluations demonstrate that their model can match (and in some cases improve upon) models which depend upon explicit dependency parse information in the input, and reliably exceed parse-free models.

I strongly vote for rejection, largely on grounds of quality, elaborated below.

Pros: The paper begins with an interesting idea and implementation, and the results support the claim that their architecture may replicate some of the contribution of dependency parse information.

Cons: The presentation is unclear on the concept of "constituent" and the motivation of the model. The later iterations of the model become rather complicated and don't seem well-motivated. The qualitative evaluations don't strongly support the claims of the paper.

Quality

1. The design of the model and the use of the word "constituent" seems conceptually problematic to me. What do you take the word "constituent" to mean for your motivation and model design? It seems to me that it might be sufficient to call the ConsTrans model a "spatial attention smoothing" model. Why isn't this a sufficient description? What does the concept "constituent" add?This question is relevant because the later model iterations are developed on further syntactic ideas (typed/labeled relations, syntactic "distance"). But if the model doesn't have a necessary syntactic framing, it's not clear these are the correct model improvements to consider.
2. The qualitative evaluations are far too light, testing only a small amount of the model performance.
  a. Grammar induction: In particular, I would appreciate a far more in-depth evaluation of the inferred constituent structures. How do they compare to gold and silver dependency parses of within-domain sentences? The current evaluation checks for model inferences on just one short span of text (the aspect term). This is probably one of the easiest terms for the model to recognize as a constituent, too, since the aspect terms are known to be constituents and have verbatim copies in the input sentence. The current evaluation is also only performed on Twitter17 --- why?
  b. Interpreting learnt relation labels: I found this evaluation extremely confusing, involving an ad-hoc dependency parsing algorithm built upon an a posteriori fact discovered in model analysis (that relation embedding L2 norm indicates inverse syntactic distance). The resulting parse in Figure 5 is almost entirely incorrect and commits many basic mistakes (for example, not linking the determiner "the" with its immediately adjacent noun). The claim about linking adjectives and nouns is not particularly interesting to me, since this is far less ambitious than the motivation of the model --- if it were, the model could have been quite a bit simpler, I think.
3. Significance results are given (thanks!) but with a strangely high significance threshold (0.15 at one point and 0.2 at another). This is not a reasonable significance threshold in my view.

Clarity

Some minor comments:

1. The constituent derivation algorithm is not clear. Eqn 3 suggests that constituent probabilities are a function of token pairs, but Algorithm 1 line 11 suggests that they can be indexed by a single token position. Is something missing from the algorithm presentation?
2. Eqn 3 first condition should be i <= j, I think.
3. Figure 4 is not a complete sentence. There's no verb. There's also no clear sentiment (aspect-based or otherwise) that we could talk about for this sentence. A different sentence could serve as both a motivating example for constituency and as a more revealing example of the model's syntactic knowledge.

## Post-rebuttal response

I have read the other reviews and the authors' responses, and do not wish to change my review. The proposed model seems quite complex with somewhat unclear conceptual motivations, and does not clearly demonstrate impressive performance gains despite the complexity. I would suggest that the authors attempt to change one of these things in a later paper, either by revisiting the model design, or task choice and evaluation (to better motivate the model).

---

> ### Author Response · Authors · 2020-11-15
> **Response to AnonReviewer2**
>
> Thank you for taking time to review the paper and for the constructive feedback provided. We address your questions and comments below: \
> \
> Question 1:
> \
> a.	What was the meaning of the word "constituent" taken to be for the your motivation and model design?\
> b.	It seems that it might be sufficient to call the ConsTrans model a "spatial attention smoothing" model. \
> \
> Response to Q1a:
> \
> The motivation of the model is to group tokens into meaningful phrases supervised by the aspect-based sentiment analysis (ABSA) task, hence the term “constituent”. Consider the example “Pork and chives dumplings in this restaurant is juicy and fresh” – the model needs to understand that the “juicy and fresh” is applied to the whole phrase “pork and chives dumplings” and not just “dumplings” alone. It is therefore necessary to group the tokens “juicy”, “and”, “fresh” as a constituent (VP) and group the tokens “pork”, “chives”, “dumplings” as a constituent (NP). Wu et al. [1] stated that “a concept of phrase dependency parsing” could benefit “[opinion] mining task” since “a lot of product features are phrases”. The task of opinion mining is similar to ours where the goal is to identify opinions for product features or aspect terms.\
> \
> Response to Q1b:
> \
> We designed our models to assign higher attention weights to tokens from the same constituent, by optimizing for the ABSA objective. Since there could be large variations in the length of constituents, this allows the model to adjust weights based on the probability of words belonging to the same constituent, rather than solely based on proximity. Definitely, the concept of spatial attention smoothing is similar to our method since tokens from the same constituent are close in proximity by nature. \
> \
> Question 2:
> \
> a.	How do they compare to gold and silver dependency parses of within-domain sentences?  The current evaluation checks for model inferences on just one short span of text (the aspect term). This is probably one of the easiest terms for the model to recognize as a constituent. \
> b.	The claim about linking adjectives and nouns is not particularly interesting since this is far less ambitious than the motivation of the model.\
> \
> Response to Q2:
> \
> We acknowledge that stating that our model is able to perform “grammar induction” generally is a far too ambitious a statement since our model was only trained for ABSA. The motivation for our proposed models is to induce the necessary syntactic bias that would be useful for ABSA. Therefore, we looked at recalling aspect terms as a way to gauge if the model was able to induce sufficient syntactic bias for ABSA.\
> In our revision, we will use “interpretability of induced constituents” rather than “grammar induction” to explain this. We have performed analysis on other datasets but did not include them in the paper due to space constraint and would also include this analysis in the revised version of our paper. \
> \
> Question 3:
> \
> Significance results are given but with a strangely high significance threshold. This is not a reasonable significance threshold. \
> \
> Response to question 3:
> \
> We acknowledge that our significance results are weak for models such as RGAT-Wang (p=0.15). However, these models use dependency parsers while ours do not. Furthermore, the threshold of BERT+BL was actually lower (p=0.1).
> The higher threshold (p=0.2) used for a significance test was for grammar induction. We will remove the claims of grammar induction in a revised version and instead, use it as an interpretability analysis of our model.\
> \
> Response to clarity comments:
> \
> We will revise the paper to address the comments on clarity. The example in Figure 4 is a complete tweet, which are sometimes not in complete sentences.\
> \
> [1] Phrase Dependency Parsing for Opinion Mining. Yuanbin Wu, Qi Zhang, Xuanjing Huang, Lide Wu. EMNLP 2009.

---

### Decision · Program_Chairs · 2021-01-07
**Final Decision**

**Decision:**

Reject

**Comment:**

The paper proposes a constituent-based transformer for aspect-based sentiment analysis. The approach allows conducting aspect-based sentiment analysis to leverage the syntactic information without pre-specified dependency parse trees.

Overall, the idea is interesting. However, all the reviewers shared the following concerns:

- Paper descriptions of methodology and experiments are not clear and require significant rewriting and reorganization.
- The proposed approach is not well-justified by the empirical study presented in the paper. Especially, a more detailed ablation study is required to justify the design.

We would suggest the authors addressing the feedback from the reviewers to improve the paper.